# Should Chronic Subretinal Hemorrhage Be Treated Surgically?

**DOI:** 10.3390/jcm14207411

**Published:** 2025-10-20

**Authors:** Wojciech Luboń, Małgorzata Luboń, Wojciech Rokicki

**Affiliations:** 1Department of Ophthalmology, Faculty of Medical Sciences, Medical University of Silesia, 40-514 Katowice, Poland; 2Department of Ophthalmology, Professor K. Gibiński University Clinical Center of the Medical University of Silesia, 40-514 Katowice, Poland; kozikowskamalg@gmail.com

**Keywords:** ocular trauma, subretinal hemorrhage, pars plana vitrectomy (PPV), vitreoretinal surgery, long-term outcome, case report

## Abstract

Closed-globe blunt ocular trauma may lead to severe ophthalmic complications, including intraocular hemorrhages, traumatic cataract, and retinopathy, the management of which remains a significant clinical challenge. We report the case of a 55-year-old male who, 15 years after right-eye trauma and unsuccessful conservative therapy, underwent surgical treatment using multistage vitreoretinal procedures. Despite an initially poor functional prognosis, a marked improvement in visual acuity was achieved, from 2.0 logMAR at baseline to 0.52 logMAR after completion of treatment. Imaging (OCT, B-scan ultrasonography) confirmed complete retinal attachment and the absence of active intra- or subretinal bleeding. This case demonstrates that, even after a long interval following trauma, vitreoretinal surgical interventions may lead to partial restoration of visual function and stabilization of ocular anatomy, underscoring the importance of individualized assessment of surgical indications in chronic post-traumatic retinal disorders.

## 1. Introduction

Ocular trauma represents an important public health issue and a serious clinical challenge. According to the World Health Organization, more than 55 million eye injuries occur worldwide each year, of which 1.6 million result in permanent blindness, while 19 million lead to varying degrees of irreversible visual impairment [1]. Epidemiological data indicate that ocular trauma is one of the leading causes of unilateral vision loss among working-age adults [2]. In developed countries, ocular injuries account for approximately 5% of all cases of blindness, whereas, in developing countries, the incidence is even higher due to limited access to specialized ophthalmic care [3].

Two main categories of ocular trauma are distinguished: open- and closed-globe injuries. Open-globe injuries include perforations, penetrating wounds, and intraocular foreign bodies, whereas closed-globe (blunt) trauma leads to intraocular mechanical damage while preserving the integrity of the ocular wall [4]. Blunt trauma results in a sudden rise in intraocular pressure, causing posterior segment damage such as vitreous hemorrhage, macular injury, retinal tears, or retinal detachment [5].

Vitreous hemorrhage is one of the most common and vision-threatening post-traumatic complications. It can obscure fundus evaluation and mask additional damage, such as retinal breaks or proliferative vitreoretinopathy (PVR) [6]. In many cases, initial management involves conservative observation, as some hemorrhages may undergo spontaneous resorption. However, persistent hemorrhage, tractional changes, or the presence of additional complications constitute indications for surgical management, typically pars plana vitrectomy (PPV) [6,7].

The timing of surgical intervention after trauma is considered a crucial prognostic factor. Both experimental and clinical studies have demonstrated that hemoglobin degradation products, particularly free iron ions, exert marked cytotoxic effects on the retinal pigment epithelium (RPE) and photoreceptors [8]. Irreversible morphological alterations may occur after only 24–48 h of subretinal blood exposure, and, after one week, progressive photoreceptor degeneration is observed [9,10]. Consequently, most authors recommend early evacuation of hemorrhagic masses within the first 7–14 days after trauma [11,12].

Surgical techniques employed in the treatment of post-traumatic hemorrhages and retinal detachments include PPV, often combined with internal limiting membrane (ILM) peeling, endolaser photocoagulation, and tamponade with gas or silicone oil [13,14,15]. The choice of technique depends on the anatomic characteristics of the lesions, the interval since trauma, and the functional prognosis. Reports indicate that outcomes may be favorable in cases of post-traumatic macular holes and giant retinal tears, particularly when comprehensive surgical techniques are applied [16,17]. Lema and Lin analyzed prognostic factors of visual acuity following surgery for open-globe injuries with retinal detachment, taking into account surgical timing, the presence of hemorrhage, and tamponade choice [12]. Ulianova et al. showed that, even in blast injuries with posterior pole involvement, vitrectomy combined with chorioretinectomy may yield satisfactory outcomes [13].

Nevertheless, some patients with severe ocular injuries, with baseline vision limited to light perception or hand movements, are often not considered for surgery due to a poor prognosis [18]. However, lack of intervention in such cases usually results in progressive retinal degeneration and complete loss of vision. Recent evidence suggests that, even in eyes with seemingly hopeless anatomic conditions, staged vitreoretinal surgery may achieve both anatomical and functional improvement [19,20].

The long-term consequences of ocular trauma extend beyond the immediate structural damage and often involve complex secondary processes within the posterior segment. Blood degradation products and inflammatory mediators promote fibrocellular membrane formation, progressive scarring, and tractional changes, often leading to proliferative vitreoretinopathy. The severity of these alterations depends not only on the type and extent of the initial injury but also on patient-related factors such as age, systemic health, and the regenerative capacity of retinal tissues. Younger individuals generally exhibit a greater potential for recovery, whereas older patients or those with comorbidities are more prone to chronic complications and irreversible vision loss. Moreover, the topography of trauma plays a critical role: injuries involving the posterior pole carry a particularly poor prognosis, while peripheral lesions may allow for partial functional preservation. These multifactorial aspects underscore the heterogeneity of outcomes in ocular trauma and highlight the importance of individualized evaluation in every case [18,19,20,21].

Of particular interest are the differences between post-traumatic hemorrhages and those associated with chronic diseases such as age-related macular degeneration (AMD) or diabetic retinopathy. Despite their dramatic clinical appearance, post-traumatic hemorrhages may show greater regenerative potential of the retina, especially in younger patients without systemic comorbidities [21,22].

The aim of this study is to present a unique case of a patient in whom surgical treatment was initiated 15 years after trauma. Despite highly unfavorable baseline factors, including chronic subretinal hemorrhage and extensive post-traumatic changes, significant improvement in functional vision was achieved. This case contributes to the discussion on patient selection for vitreoretinal surgery and emphasizes that the time interval since trauma should not be considered an absolute disqualifying factor.

## 2. Case Presentation

A 55-year-old male was referred to the Department of Ophthalmology, University Clinical Center in Katowice, for a routine ophthalmic examination. During the visit, he reported a blunt trauma to the right eye that had occurred 15 years earlier. The injury was sustained during daily activities as a result of accidental finger trauma by a child. Immediately after the event, the patient noticed a sudden, profound deterioration of vision in the affected eye. Visual function was reduced to hand motion perception with light perception and correct light projection (2.0 logMAR, ~0.01 decimal).

### 2.1. Functioning Prior to Treatment

Since the injury, the patient had functioned essentially monocularly. The left eye remained fully functional, with visual acuity of 1.0 (0.0 logMAR), enabling daily activities and continuation of occupational duties. However, the patient reported:Difficulty with depth perception and spatial orientation,Impairments in tasks requiring precise binocular coordination (e.g., manual work, cycling),Recurrent headaches related to overuse of the left eye,Inability to continue his previous profession, which required good stereoscopic vision (he had previously worked as an assembler).

Although the patient adapted to monocular functioning, he reported a significant reduction in quality of life.

### 2.2. Baseline Examination

Ophthalmic examination of the right eye revealed:Anterior segment—inferior corneal scar, partial post-traumatic sectoral cataract,Vitreous—presence of dense, chronic hemorrhagic remnants,Retina—multiple intraretinal hemorrhages, signs of post-traumatic retinopathy.

#### Imaging Studies


Optical coherence tomography (OCT): thinning of the outer retinal layers in the macula, presence of submacular blood remnants, disrupted photoreceptor architecture, and discontinuity of the IS/OS junction; vitreomacular tractional changes within the inner retinal layers.B-scan ultrasonography: dense echogenic material corresponding to vitreous hemorrhage, irregular and undulating retinal attachment in the posterior pole suggestive of tractional component.Fluorescein angiography (FA): leakage from perifoveal vessels, areas of hypofluorescence corresponding to blood clots, and hyperfluorescent foci consistent with scarring.


The left eye remained entirely healthy, with normal OCT and FA findings, confirming compensatory monocular functioning, with distance visual acuity of 0.0 logMAR (1.0 Snellen).

### 2.3. Therapeutic Decision

Due to the absence of hemorrhage resorption and suspicion of complex vitreoretinal pathology, the patient was scheduled for planned staged vitreoretinal surgery. Between May and November 2024, three surgical procedures were performed on the right eye, with techniques tailored to the evolving clinical situation.

#### 2.3.1. Surgical Interventions

🔹 First surgery—23 May 2024

Pars plana vitrectomy (PPV) was performed. Internal limiting membrane (ILM) peeling in the macular area, aspiration of chronic subretinal hemorrhagic remnants, and focal endophotocoagulation of the retina were carried out. A 25% SF_6_ gas tamponade was applied. A 25% SF_6_ mixture was selected to enhance the pneumatic retinopexy effect and to provide more stable retinal tamponade, thereby minimizing the risk of postoperative hypotony. The intraoperative image of the vitreous chamber and the fundus of the eye is shown in Figure 1

Postoperative course: Two weeks postoperatively, partial improvement in visual axis clarity was achieved, although functional acuity remained low (object recognition, but no reading improvement). OCT showed partial clearance of the submacular space and limited re-approximation of photoreceptor architecture.

🔹 Second surgery—1 August 2024

At follow-up, B-scan ultrasonography revealed retinal detachment with a macular tear. Repeat PPV with perfluorocarbon liquid (perfluorodecalin), extended retinal endolaser photocoagulation, and cataract extraction with intraocular lens implantation (RayOne Aspheric + 23.5 D) were performed. Tamponade with 1000 cSt silicone oil was applied. The procedure resulted in complete retinal reattachment.

Postoperative course: The retina remained fully attached postoperatively. OCT confirmed reattachment and partial restoration of inner retinal layers. Visual acuity was 1.6 logMAR.

🔹 Third surgery—7 November 2024

After three months of stability, silicone oil removal was performed with final gas tamponade using 25% SF_6_. The procedure was uneventful.

Postoperative course: The retina remained permanently attached. OCT showed no new hemorrhages or vitreoretinal tractions. Visual acuity gradually improved in the following weeks.

#### 2.3.2. Perioperative and Pharmacological Management

Standard postoperative pharmacological treatment included topical dexamethasone (6×/day), levofloxacin (Oftaquix, 6×/day), topical hyperosmotic anti-edematous eye drops (Cornesin 3×/day) to reduce corneal edema, and systemic dexamethasone (Dexaven 8 mg/day during hospitalization). Thromboprophylaxis was provided according to internal medicine recommendations (enoxaparin sodium, Clexane 40 mg s.c. once daily).

In the early postoperative period, intraocular pressure (IOP) was closely monitored to ensure stable gas expansion and prevent pressure-related complications.

### 2.4. Final Outcome and Long-Term Follow-Up

Seven months after the first surgery:Right eye visual acuity—0.3 Snellen (0.52 logMAR),OCT—complete macular reattachment, thinning of photoreceptor layers, no active hemorrhage,B-scan—no traction, no retinal detachment,FA—no active leakage.

The postoperative image of the patient’s ocular fundus after completion of treatment is shown in Figure 2.

#### Final Effect

Following completion of surgical treatment and recovery, final right eye visual acuity was 0.3 Snellen (0.5 logMAR), representing a marked improvement compared to baseline (hand motion perception only, 2.0 logMAR). This provided the patient with a functionally and socially meaningful level of vision.

### 2.5. Case Summary

A patient who had functioned monocularly for 15 years regained functional visual acuity in the traumatized eye through staged vitreoretinal surgery. This case demonstrates that, even after a long interval following trauma, surgical intervention may restore anatomic retinal integrity and lead to visual improvement.

## 3. Discussion

### 3.1. Pathophysiology of Subretinal and Vitreous Hemorrhage

Extravasation of blood into intraocular spaces, particularly the vitreous cavity and subretinal space, represents one of the most destructive sequelae of ocular trauma. While vitreous hemorrhage may in some cases resolve spontaneously or regress with conservative management, intra-, sub-, and preretinal hemorrhages are associated with a significantly poorer prognosis [3,14,15,16]. The toxic mechanisms associated with intraocular blood include:Mechanical effects: blood mass displaces and detaches the retina, exerting pressure on the outer photoreceptor segments and impairing their metabolic function.Biochemical effects: hemoglobin degradation products, particularly free iron ions, promote the generation of reactive oxygen species (ROS), lipid peroxidation, and activation of apoptotic pathways, leading to photoreceptor and retinal pigment epithelium (RPE) cell degeneration [15,16,17,18].Inflammatory effects: the presence of blood in the subretinal space induces microglial activation and macrophage migration, contributing to fibrosis and proliferative vitreoretinopathy (PVR) [19].

Experimental studies have shown that photoreceptor degeneration begins as early as 24–48 h after exposure [20]. In the following days, irreversible morphological alterations occur, and within weeks, RPE atrophy and complete macular disintegration ensue [21]. The absence of physiological drainage of subretinal blood further enhances toxicity, leading to irreversible macular damage [19,21]. For this reason, most authors advocate early surgical evacuation of subretinal blood [22]. Prognosis also depends on the hemorrhage location. Comparative studies have demonstrated that subretinal hemorrhage carries a far worse prognosis than intraretinal hemorrhage, mainly due to the direct contact of hemoglobin breakdown products with outer photoreceptors and RPE cells [22,23]. Although intraretinal hemorrhage may also compromise retinal function through mechanical compression and localized ischemia, it usually does not cause such rapid and irreversible photoreceptor degeneration as subretinal hemorrhage [24].

Experimental studies by Glatt and Machemer [16] and by Hope and Dawson [17] have demonstrated that early hemoglobin degradation products induce rapid photoreceptor necrosis and RPE atrophy through oxidative stress and iron-mediated cytotoxicity. However, over time, the biochemical profile of residual subretinal blood may change, with transformation of free hemoglobin into less reactive iron-storage complexes such as hemosiderin and ferritin. These deposits exert a lower degree of oxidative stress, potentially allowing partial preservation of outer retinal elements. In addition, glial and RPE remodeling may contribute to limited structural stability and residual function even after prolonged injury. This adaptive response could explain why, in rare cases such as the present one, measurable visual recovery remains possible despite long-standing exposure to subretinal hemorrhage.

### 3.2. The Role of Exposure Time

The literature emphasizes the critical role of time from trauma to surgical intervention. The best outcomes are achieved when treatment is initiated within the first days to weeks [23]. The critical period is generally considered to be 7–14 days, after which the chance of functional macular recovery declines dramatically [24].

In this context, the present case is exceptional: intervention was undertaken 15 years after trauma. According to current knowledge, such prolonged subretinal blood exposure should irreversibly damage retinal receptor elements. The partial resilience of the patient’s retina to the toxic effects of chronic hemorrhage was therefore unexpected. The final visual acuity achieved not only challenges expectations but also raises questions about whether timing should be regarded as an absolute contraindication to vitreoretinal surgery.

In this case, functional improvement suggests that:Biochemical transformation of blood remnants—in the subretinal space, which lacks an active phagocytic system, hemoglobin breakdown products may gradually transform into less reactive forms (e.g., hemosiderin, ferritin) [25]. In the present case, after more than a decade, primary hemoglobin toxicity may have subsided, leaving less harmful deposits.Patient-related factors—at the time of trauma, the patient was approximately 40 years old (relatively young), which may have supported partial functional reserve and neuronal plasticity, enhancing the potential for retinal reorganization [26].Absence of systemic comorbidities (e.g., diabetes, hypertension) limited additional damaging influences and may have preserved regenerative capacity [27].

### 3.3. Comparison with the Literature

Reports of delayed surgical interventions in such cases are scarce.

Chen et al. [10] described massive subretinal hemorrhages where early PPV combined with tPA led to recovery of visual acuity up to 0.4 Snellen; however, interventions were performed within days to weeks, not years.Boral et al. [22] presented cases of retinal hemorrhage treated surgically using various techniques, showing that earlier intervention yielded superior functional outcomes compared to delayed management.Wu et al. [9] demonstrated that early surgical intervention in closed-globe injuries with massive vitreous hemorrhage reduces the risk of PVR and improves retinal survival.Lema and Lin [12] identified surgical timing as a key prognostic factor in open-globe injuries with retinal detachment.

Against this background, the present case is highly unusual; functional improvement was achieved despite an extraordinarily long interval since trauma.

The etiology of retinal hemorrhage plays a crucial prognostic role. Hemorrhages may arise from neovascular complications (e.g., AMD), systemic diseases (e.g., hypertension, diabetes, coagulopathy), or acute trauma. Several studies indicate that trauma-related hemorrhages, despite their dramatic clinical course, may yield relatively better functional outcomes in selected patients if retinal anatomy remains partially preserved [28]. Retinal regenerative mechanisms are more effective in younger patients with isolated trauma than in older patients with chronic vascular or metabolic disease [29].

In contrast, spontaneous subretinal hemorrhages, particularly those related to AMD or choroidopathy, tend to spread and rapidly damage the RPE and photoreceptors, with limited regenerative potential due to underlying degenerative changes [30]. Similarly, in systemic diseases such as hypertension or coagulation disorders, systemic factors—microangiopathy, impaired retinal perfusion, and altered inflammatory responses—further worsen outcomes [31].

Recent studies addressing submacular hemorrhage secondary to age-related macular degeneration (AMD) have shown that early pars plana vitrectomy combined with recombinant tissue plasminogen activator (rtPA) and gas or silicone oil tamponade may lead to partial visual improvement when performed within days after symptom onset [30]. However, functional outcomes in such cases are typically limited by underlying degenerative changes of the retinal pigment epithelium and photoreceptors, which reduce the regenerative potential of the retina. In contrast, in trauma-related hemorrhage, the absence of chronic vascular pathology and preserved microcirculation may allow for greater recovery once toxic blood degradation products are removed. The present case differs from AMD-associated cases not only in etiology and chronicity but also in the unexpectedly favorable functional outcome despite extreme delay in intervention. These differences further support the hypothesis that, under selected conditions, surgical removal of chronic hemorrhagic material can restore partial visual function even after a prolonged period.

In the present case, the hemorrhage resulted from a single blunt injury in a patient without systemic comorbidities, likely allowing residual retinal function to persist despite prolonged exposure to hemorrhagic material. This may explain the unexpectedly favorable functional response to surgery, despite a theoretically poor anatomical baseline. Notably, OCT evaluation revealed preserved retinal architecture in certain layers, which encouraged the surgical attempt to remove hemorrhagic products displacing the macula.

### 3.4. Surgical Strategies and Choice of Tamponade

Various surgical approaches are used in post-traumatic hemorrhage and retinal detachment. Key elements include:Pars plana vitrectomy (PPV): removal of hemorrhagic remnants,ILM peeling: reducing traction and promoting retinal integration,Endolaser photocoagulation: securing retinal tears and degenerative foci,Gas tamponade (SF_6_, C_3_F_8_): applied in localized cases, providing temporary retinal apposition,Silicone oil tamponade: preferred in complex detachments with high recurrence risk [28,29].

In this case, both tamponade methods were required. Initial gas tamponade was insufficient, and silicone oil was necessary for durable reattachment. This raises the question of whether silicone oil should be considered as a primary tamponade in long-standing submacular hemorrhages with macular involvement and posterior retinal tears. Currently, no consensus exists regarding tamponade choice in traumatic cases; decisions should be individualized, depending on lesion morphology and the feasibility of stable retinal reattachment [32,33,34,35].

### 3.5. Visual Rehabilitation

Visual rehabilitation is a critical aspect of post-traumatic sequelae. Patients who have functioned monocularly for years often experience difficulty readapting to binocular vision once partial function is regained. Rehabilitation strategies include:Central and eccentric fixation training,Accommodative exercises,Adaptation for reading and use of optical aids.

The final improvement—visual acuity from 2.0 logMAR to 0.52 logMAR—was not only surprising but also clinically meaningful. Previous studies have shown that retinal exposure to blood breakdown products longer than 7–14 days significantly reduces the chance of macular functional recovery [16,17,21]. In this patient, partial functional reserve and preserved photoreceptor structures may have enabled positive response to surgery once the toxic factor was eliminated. Although functional vision was restored, stereoscopic vision remained limited.

In addition to the measurable improvement in visual acuity, the patient experienced a significant enhancement in daily visual function. After regaining partial vision in the affected eye, he was able to resume reading large-print text and perform basic tasks requiring peripheral depth perception, such as navigating stairs and recognizing objects in motion. Visual readaptation was supported by structured training of central and eccentric fixation, as well as by the use of individualized optical aids for near vision. The patient also reported a reduction in asthenopic symptoms and improved spatial orientation, contributing to better social and occupational functioning. These findings underscore the importance of postoperative rehabilitation and functional assessment in evaluating the overall benefit of late surgical intervention.

### 3.6. Clinical and Research Implications

This case raises several practical and scientific questions:Should surgical eligibility criteria be more liberal? Traditionally, patients with long-standing hemorrhage were disqualified. This case suggests reconsideration, even after years.Which mechanisms underlie preservation of partial retinal functional reserve? Histopathological and experimental studies are needed to clarify the effects of chronic intraocular blood.Which tamponade is optimal in chronic cases? The lack of consensus highlights the need for prospective studies.Can visual rehabilitation protocols be developed for patients with late visual recovery?

### 3.7. Discussion Summary

This case represents a rare example of successful late surgical management of chronic post-traumatic subretinal hemorrhage. The findings suggest that, in selected patients, vitreoretinal surgery may help restore partial anatomical integrity and improve visual function even after a long interval following trauma. The outcome supports the view, reported in previous studies [12,22,28], that the potential for functional recovery depends not only on the time elapsed since injury but also on the morphological preservation of the retina and the absence of systemic comorbidities. However, these observations should be interpreted with caution given the single-case nature of the report and the potential influence of individual patient factors.

### 3.8. Limitations

This study presents a single clinical case; therefore, the conclusions should be interpreted with caution. The observations described here reflect an individual outcome and cannot be generalized to all patients with chronic post-traumatic subretinal hemorrhage. Additional studies involving larger patient cohorts are required to determine whether similar anatomical and functional outcomes can be consistently reproduced.

## 4. Conclusions

The present case illustrates that delayed surgical intervention in chronic post-traumatic subretinal hemorrhage can, under specific conditions, lead to meaningful anatomical and functional improvement. The findings may indicate that time since trauma should not be regarded as an absolute contraindication to surgery when multimodal imaging reveals preserved retinal architecture and potential for recovery. Careful morphological assessment using optical coherence tomography, fluorescein angiography, and ultrasonography is therefore essential for selecting candidates who may benefit from surgery despite an extended interval after injury.

A staged surgical approach combining pars plana vitrectomy, internal limiting membrane peeling, endolaser photocoagulation, and appropriate tamponade appears to provide stable anatomical outcomes in such chronic cases. Moreover, this case highlights the need for further investigation into the cellular and biochemical mechanisms that enable partial preservation of retinal function after prolonged blood exposure. Although the conclusions cannot be generalized from a single observation, the results contribute to the ongoing discussion on surgical eligibility criteria and the potential role of late vitreoretinal intervention in chronic post-traumatic retinal disorders.

## Figures and Tables

**Figure 1 jcm-14-07411-f001:**
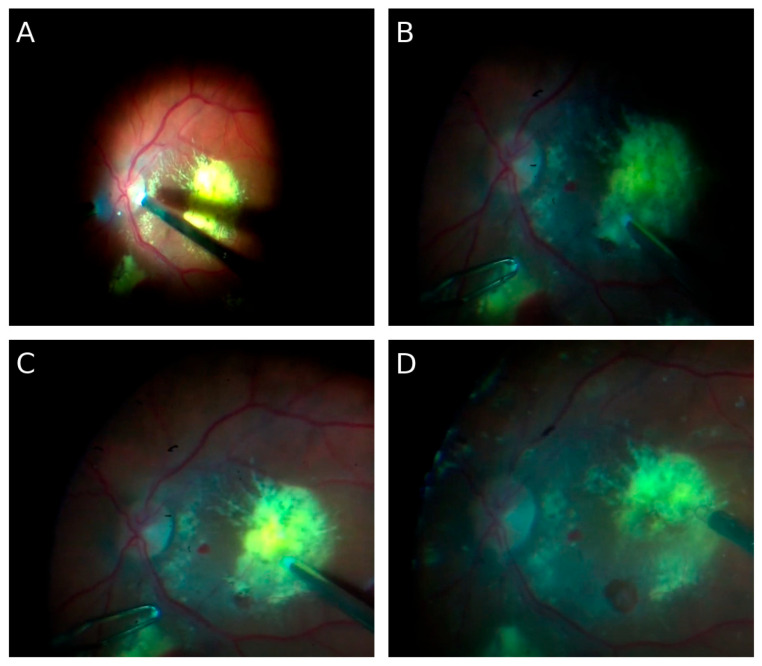
(**A**–**D**) Intraoperative images of the ocular fundus. Representative intraoperative views showing extensive pathological changes in the ocular fundus of the patient.

**Figure 2 jcm-14-07411-f002:**
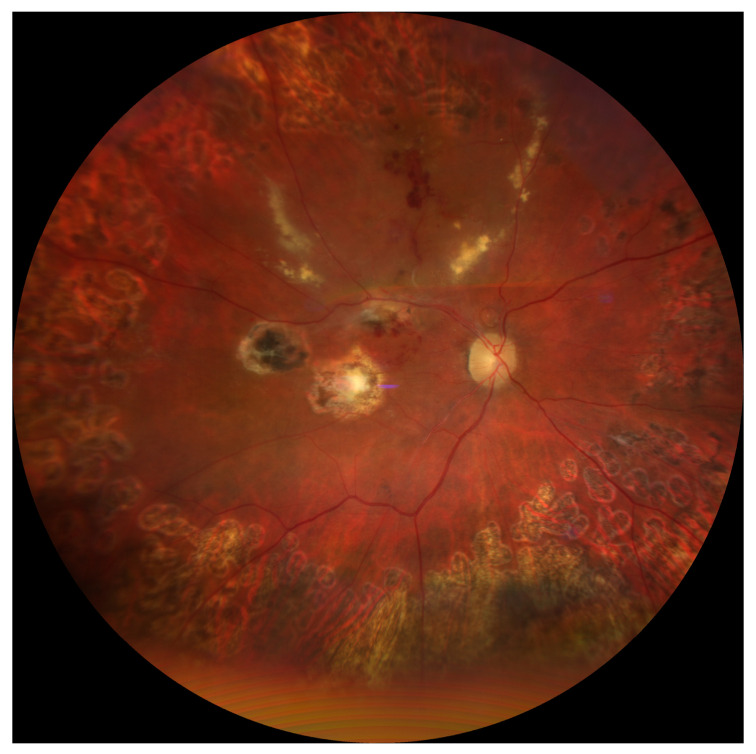
Post-treatment image of the ocular fundus. Representative fundus image of the patient after completion of therapy, demonstrating the postoperative appearance of the retina.

## Data Availability

The data used to support the findings of this study are available from the corresponding author upon request.

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
