# Peer review of "Should Chronic Subretinal Hemorrhage Be Treated Surgically?"

_jcm, 2025, doi:10.3390/jcm14207411_

Round 1

Reviewer 1 Report

Comments and Suggestions for Authors

This case report uniquely addresses the question of whether chronic subretinal/vitreous hemorrhage following blunt trauma can be surgically treated at a very late stage (after 15 years).

Clearly Emphasizing Limitations

Because the study was conducted on a single case, its generalizability is limited. This should be stated more clearly and firmly in the Limitations section.

Deepening the Pathophysiological Discussion

The long-term toxic effects of subretinal hemorrhage on photoreceptors and RPE have been demonstrated in animal studies in the literature (e.g., Glatt & Machemer, Hope & Dawson). In light of these data, a more comprehensive discussion of how partial functional reserve can be preserved even after 15 years would be beneficial.

Detailing Functional Rehabilitation

Beyond visual acuity improvement, adding information on the patient's daily life functions (reading, mobility, social adaptation) and, if necessary, low vision rehabilitation (eccentric fixation, optical aids) will enhance the clinical value of the article.

Strengthening Literature Links

The study's findings should be compared with those in recent publications (especially vitrectomy/rtPA/gas or silicone oil treatments for submacular hemorrhages due to AMD), and similarities/differences should be discussed. This will help position the case better in the literature.

Revising Language and Style

In the discussion, it is recommended that conclusions be presented in a more neutral, literature-based style. Instead of broad generalizations, the clinical message of the case should be emphasized, and also reminding of the limitations would be helpful.

Author Response

Comments 1: Because the study was conducted on a single case, its generalizability is limited. This should be stated more clearly and firmly in the Limitations section.

Response 1: I fully agree with the reviewer that the generalizability of my observations is limited because this report describes a single case. I have now added a dedicated paragraph under the Limitations section to emphasize this aspect more explicitly. The revised text clearly states that the conclusions should be interpreted with caution and cannot be extrapolated to the general population of patients with chronic post-traumatic subretinal hemorrhage.
Page 12, lines 396-401

Comments 2: The long-term toxic effects of subretinal hemorrhage on photoreceptors and RPE have been demonstrated in animal studies in the literature (e.g., Glatt & Machemer, Hope & Dawson). In light of these data, a more comprehensive discussion of how partial functional reserve can be preserved even after 15 years would be beneficial.

Response 2: I appreciate the reviewer’s insightful comment and fully agree that a more detailed discussion of the potential mechanisms responsible for the preservation of partial functional reserve after long-term subretinal hemorrhage would strengthen the manuscript. I have therefore expanded the pathophysiological section to include a concise synthesis of experimental data (Glatt & Machemer, Hope & Dawson) and recent hypotheses explaining why limited retinal function may persist despite prolonged blood exposure. This addition clarifies how chronic biochemical adaptation and cellular remodeling might account for the functional improvement observed in this patient.
Page 8, lines 250-259

Comments 3: Beyond visual acuity improvement, adding information on the patient's daily life functions (reading, mobility, social adaptation) and, if necessary, low vision rehabilitation (eccentric fixation, optical aids) will enhance the clinical value of the article.

Response 3: I thank the reviewer for this valuable suggestion. I agree that the description of postoperative rehabilitation and functional adaptation adds clinical relevance to the case. I have therefore expanded the Visual Rehabilitation section to include a concise summary of the patient’s functional outcomes and the low-vision rehabilitation strategies that supported his reintegration into daily activities. This addition highlights not only the visual acuity improvement but also the meaningful enhancement in quality of life achieved after long-term monocular functioning.
page 11, lines 362-371

Comments 4: The study's findings should be compared with those in recent publications (especially vitrectomy/rtPA/gas or silicone oil treatments for submacular hemorrhages due to AMD), and similarities/differences should be discussed. This will help position the case better in the literature.

Respone 4: I appreciate the reviewer’s suggestion to further contextualize the present case by comparing it with recent reports of vitrectomy and submacular hemorrhage management related to age-related macular degeneration (AMD). I have therefore expanded the Comparison with the Literature section to include this aspect. The new paragraph summarizes outcomes from rtPA-assisted PPV and gas or silicone oil tamponade in AMD-associated cases, emphasizing the distinct pathophysiology and regenerative potential that differentiate post-traumatic from degenerative subretinal hemorrhages. 
Page 10, lines 310-322

Comments 5: In the discussion, it is recommended that conclusions be presented in a more neutral, literature-based style. Instead of broad generalizations, the clinical message of the case should be emphasized, and also reminding of the limitations would be helpful.

Response 5: I appreciate the reviewer’s valuable suggestion regarding the tone and style of the Discussion and Conclusions. I have revised these sections to ensure a more neutral and literature-based formulation. Overly generalized statements have been replaced with evidence-oriented phrasing, and additional references to existing studies are now included where appropriate. The revised text emphasizes the clinical implications of the case while maintaining a cautious interpretation consistent with the study’s single-patient nature.
Page 11, lines 386-394
Page 12, lines 404-419

Reviewer 2 Report

Comments and Suggestions for Authors

The manuscript presents a case report of a patient with long-term, untreated post-traumatic ocular damage, demonstrating that functional vision can still be restored through vitreoretinal surgery. This study emphasizes that while the time elapsed since trauma can serve as a reference point, it should not be the sole determinant for surgical eligibility.

Although the patient experienced a remarkable visual recovery following surgery 15 years after the initial trauma, the authors thoughtfully consider additional factors—such as patient age, severity of the trauma, and systemic comorbidities—that may have contributed to this unexpected outcome.

The report provides an important perspective on surgical decision-making, suggesting that criteria for eligibility should be assessed with greater nuance. Specifically, less emphasis should be placed solely on the timing of surgery after trauma, and more consideration should be given to the potential for recovery, the extent of injury, and overall patient health.

Overall, this is a compelling case report that not only documents an exceptional recovery but also offers valuable insight into refining surgical criteria to better serve patients with chronic post-traumatic ocular conditions.

Minor typo:

Line 13 – The word “includeing” should be corrected to “including.”

Additionally, I would strongly recommend that the authors carefully proofread the manuscript for consistent and appropriate use of terminology throughout.

Author Response

Comments 1: Line 13 – The word “includeing” should be corrected to “including.”

Response 1: I thank the reviewer for identifying this typographical error. I have corrected “includeing” to “including” in line 13 of the manuscript.
Page 1, lines 13-14

Comments 2: I would strongly recommend that the authors carefully proofread the manuscript for consistent and appropriate use of terminology throughout.

Response 2: I thank the reviewer for the valuable remark regarding consistency and appropriate use of terminology. I have carefully proofread the entire manuscript and made minor stylistic and terminological corrections to ensure coherence and alignment with the Journal of Clinical Medicine style. The adjustments include refinement of neutral tone (“may help restore” instead of “may restore”), standardization of ophthalmic terminology (“subretinal hemorrhage” instead of “submacular”). The manuscript was professionally proofread for English and terminology consistency,
Page 1, lines 13 and 22,
Page 4, line 114,
Page 5, line 155 and 175
Page 8, line 227
Page 9, line 269

Comments 3: The manuscript presents a compelling case and offers valuable insight into refining surgical criteria for chronic post-traumatic ocular conditions.

Response 3: I sincerely thank the reviewer for this encouraging evaluation and for recognizing the clinical significance of this case. I am pleased that the revisions successfully clarified the rationale for nuanced surgical decision-making in chronic post-traumatic subretinal hemorrhage. I appreciate the reviewer’s thoughtful summary and positive feedback.

Round 2

Reviewer 1 Report

Comments and Suggestions for Authors

Thanks for the implementing of the suggestions.

Author Response

I sincerely thank the Editor for the positive feedback and for the careful review of my revisions. I greatly appreciate your time and consideration.